# A Literature Review of Changes in Phase II Drug-Metabolizing Enzyme and Drug Transporter Expression during Pregnancy

**DOI:** 10.3390/pharmaceutics15112624

**Published:** 2023-11-15

**Authors:** Christine Gong, Lynn N. Bertagnolli, David W. Boulton, Paola Coppola

**Affiliations:** 1School of Pharmacy, University of Pittsburgh, Pittsburgh, PA 15261, USA; 2AstraZeneca LP, Biopharmaceuticals R&D, Clinical Pharmacology & Safety Sciences, Clinical Pharmacology & Quantitative Pharmacology, Gaithersburg, MD 20878, USA; 3AstraZeneca LP, Biopharmaceuticals R&D, Clinical Pharmacology & Safety Sciences, Clinical Pharmacology & Quantitative Pharmacology, Cambridge CB2 0AA, UK

**Keywords:** pregnancy, placenta, gestational change, phase II enzyme, drug transporter, physiologically based pharmacokinetic (PBPK) modeling

## Abstract

The purpose of this literature review is to comprehensively summarize changes in the expression of phase II drug-metabolizing enzymes and drug transporters in both the pregnant woman and the placenta. Using PubMed^®^, a systematic search was conducted to identify literature relevant to drug metabolism and transport in pregnancy. PubMed was searched with pre-specified terms during the period of 26 May 2023 to 10 July 2023. The final dataset of 142 manuscripts was evaluated for evidence regarding the effect of gestational age and hormonal regulation on the expression of phase II enzymes (*n* = 16) and drug transporters (*n* = 38) in the pregnant woman and in the placenta. This comprehensive review exposes gaps in current knowledge of phase II enzyme and drug transporter localization, expression, and regulation during pregnancy, which emphasizes the need for further research. Moreover, the information collected in this review regarding phase II drug-metabolizing enzyme and drug transporter changes will aid in optimizing pregnancy physiologically based pharmacokinetic (PBPK) models to inform dose selection in the pregnant population.

## 1. Introduction

Physiological changes during pregnancy affect drug pharmacokinetics (PK), including absorption, distribution, metabolism, and elimination [1]. PK changes that affect the activity of drug-metabolizing enzymes and drug transporters can differ in each pregnancy trimester [2]. These gestational changes inform the selection of safe and effective drug doses for pregnant patients and guide the decision to conduct appropriate dose monitoring during pregnancy. Approximately 81% of pregnant women take at least one prescription or over-the-counter medication during gestation, excluding vitamins and dietary supplements [3]. Despite the high prevalence of medication use during pregnancy, most medications are administered “off-label” to pregnant patients, with doses based on PK data from nonpregnant individuals [4]. With limited clinical trials conducted in the pregnant population, PK changes during pregnancy are poorly characterized and optimal dose regimens for pregnant patients are insufficiently investigated [5].

Physiologically based pharmacokinetic (PBPK) modeling is an increasingly used method for predicting drug exposure during pregnancy [5]. Utilizing mathematical equations, PBPK models incorporate known physiological changes into a mechanistic model that describes drug PK [6]. Informed pregnancy PBPK models may support the evaluation of PK data in the pregnant population, guide the proposal of safe and effective doses for clinical drug development programs, and supplement clinical pharmacology studies during regulatory approval [7]. One aspect of PBPK model predictions is informed by adequate knowledge of gestational changes in drug-metabolizing enzymes and drug transporters; for instance, knowledge of these gestational changes may inform the robust prediction of drug renal clearance, systemic exposure, and their changes across pregnancy trimesters [6]. However, due to sparse or conflicting data, the gestational changes of only a small number of phase II enzymes and drug transporters have been incorporated into PBPK modeling software. The current gaps in knowledge emphasize the need to study changes in phase II enzymes and drug transporters across gestational trimesters.

Several literature reviews have been published that examine changes in enzyme and/or transporter expression in the pregnant woman and in the placenta. Gestational changes in select phase II enzymes and renal drug transporters have been elucidated through pharmacokinetic analysis of probe drugs administered during pregnancy [8,9]. Additional evidence suggests that drug-metabolizing enzymes and drug transporters in the placenta largely affect fetal drug exposure [10], but the lack of available or consistent information regarding gestational changes in some placental transporters necessitates further research [11]. Transcription factors, steroid hormones, genetic variations, and pregnancy complications have also been observed to change the expression of placental drug transporters [12,13]. Though abundant evidence describes gestational changes in enzyme and transporter expression, to our knowledge, the evidence and its implications to PBPK modeling have not been synthesized into a single manuscript. 

The purpose of this literature review is to comprehensively summarize changes in the expression or activity of phase II enzymes and drug transporters in both the pregnant woman (e.g., intestinal, renal, or hepatic enzymes and transporters) and the placenta, as currently described in the literature. Where possible, gestational changes are detailed across the three trimesters of pregnancy and postpartum. The effects of hormonal regulation on enzyme or transporter activity are also included in this review. 

This comprehensive review aims to expand upon the existing knowledge of drug metabolism and transport during pregnancy, aid incorporation of enzyme and transporter changes into PBPK models, and inform predictions of PK changes in the pregnant population. Though more research is necessary to understand the complete landscape of gestational changes in phase II enzymes and drug transporters, the data collected in this review can be utilized to optimize pregnancy PBPK models, especially as Sponsors and Regulators seek to serve this population in drug development programs.

## 2. Placenta

The human placenta links the fetus to the mother, providing nutrients to and removing wastes from the fetal circulation [14]. In addition to its function of supporting fetal metabolism, the placenta may play a fetoprotective role by extruding xenobiotics, such as drugs, from the fetal circulation.

### 2.1. Placental Anatomy

The human placenta possesses a hemochorial structure, in which the fetal tissue directly contacts maternal blood [15]. The fetal tissue consists of syncytiotrophoblasts, cytotrophoblasts, and vascular endothelial cells, of which the syncytiotrophoblast cells are the main barrier to drug transport [14,15]. The syncytiotrophoblasts comprise a maternal-facing brush border membrane (i.e., apical membrane) and a fetal-facing basal membrane (i.e., basolateral membrane) [14]. The apical membrane constitutes the main site of exchange for drugs, nutrients, and endogenous molecules between the maternal and fetal circulation, while the basolateral membrane provides structural attachment to cytotrophoblasts or fetal connective tissue, which houses the fetal capillaries [14,16]. Molecules are transported from the maternal uterine vasculature, across the apical and basolateral membranes of the syncytiotrophoblasts, through the fetal endothelium, and to the fetal circulation [1].

### 2.2. Placental Drug Transport

Both passive diffusion and transporter-mediated transfer are involved in the transport of drug molecules across the placental syncytiotrophoblast [16]. The rate of passive diffusion of a drug can be affected by its molecular weight, pKa, and/or lipophilicity [17]. In general, drugs of high molecular weight demonstrate limited passive transport across the placenta. Drugs that are unionized at physiological pH tend to diffuse across the placental membrane more easily than drugs that are ionized. Passive diffusion is also a common transport mechanism for lipophilic drugs [16]. 

Drug transporters are membrane proteins that efflux or influx endogenous and exogenous substances [15], offering an alternative mechanism of transport for drug substrates that do not easily diffuse across the placental syncytiotrophoblast [16]. For substrates of multiple drug transporters, the net direction of transport is determined by the relative abundance of each transporter, the affinity of the drug for each transporter, and the mechanisms that regulate transporter activity [14]. The localization of select placental drug transporters and the directionality of their transport are illustrated in Figure 1. A description of each drug transporter family is detailed in subsequent sections.

## 3. Drug Transporters

Although drug transporters facilitate the passage of both endogenous and exogenous substrates across cellular membranes, in this review, we focus on the role of drug transporters in the transport of drug substrates. As important contributors to drug PK, drug transporters participate in drug absorption in the gastrointestinal tract; drug distribution to various organs, such as the placenta; and drug excretion by the liver and kidney [15]. Drug transporters are classified into two superfamilies: ATP-binding cassette (ABC) transporters and solute carrier (SLC) transporters.

### 3.1. ATP-Binding Cassette Superfamily

Transporters belonging to the ABC superfamily utilize ATP hydrolysis to efflux drugs from cells [15,37]. ABC transporters participate in the final steps of drug PK [38], including drug excretion by the liver or kidney and drug secretion by the placenta.

#### 3.1.1. Multidrug Resistance Protein (MDR) Family

MDR1, alternatively known as P-glycoprotein (P-gp), is the most widely studied transporter within the MDR protein family (*ABCB* gene family) [10]. Located in the liver, kidney, intestine, and brain, MDR transporters facilitate the secretion of substrates from cells [15]. The placental MDR1, MDR3, and bile salt export pump (BSEP) may limit fetal exposure to toxic and nontoxic compounds [39,40].

#### 3.1.2. Multidrug Resistance-Associated Protein (MRP) Family

Transporters in the MRP protein family (*ABCC* gene family) are expressed in various tissues, including the liver, kidney, intestine, and brain [41]. The MRP transporters export anionic drugs, as well as their sulfate, glucuronate, and glutathione metabolites. Placental MRP1 [14], MRP2 [22], MRP3 [23], and MRP5 [25] also efflux anionic and/or conjugated substrates, and the direction of transport to or from the fetal circulation depends on the transporter’s localization in the basolateral or apical membrane (Figure 1) [14].

#### 3.1.3. Breast Cancer Resistance Protein (BCRP)

BCRP belongs to the *ABCG* gene family, with protein expression identified in the liver, kidney, intestine, and lung [14]. Though it can transport charged drugs and glucuronated metabolites [14], BCRP demonstrates a preference for the transport of sulfate conjugates [42]. In the placenta, BCRP effluxes xenobiotics [26], sulfated steroids [29], and bile acids [43] from the fetal circulation, which may confer protection to the fetus [42].

### 3.2. Solute Carrier Superfamily

Transporters belonging to the SLC superfamily influx drugs into cells via secondary active transport (e.g., antiport or symport) and passive transport (e.g., uniport) [37,38]. SLC transporters enable various processes of drug PK [38], such as drug absorption across the luminal membrane of enterocytes and drug distribution across the apical membrane of placental syncytiotrophoblasts.

#### 3.2.1. Organic Anion-Transporting Polypeptide (OATP) Family

The OATP transporters, encoded by the *SLC21/SLCO* genes, are expressed in the liver, kidney, intestine, and brain [44]. OATP substrates include amphipathic organic compounds, such as drugs, bile acids, steroids, hormones, and peptides. OATP-mediated uptake may work in concert with ABC-mediated efflux [44], an interaction that has also been observed between placental transporters, including OATP2B1 and BCRP [29].

#### 3.2.2. Organic Cation Transporter (OCT) Family

Transporters in the OCT family are encoded by the *SLC22* genes [16]. OCT1 is largely expressed in the liver, OCT2 in the kidney, and OCT3 in the central nervous system and placenta; there is minimal overlap in tissue distribution between members of the OCT family [45]. OCTs enable the passive uptake of organic cations, which is coupled with cellular efflux facilitated by a different transporter on the opposite cellular membrane [16]. In addition to transporting cationic compounds, placental OCT3 also exhibits a high affinity for monoamines [30].

#### 3.2.3. Organic Cation/Carnitine Transporter (OCTN) Family

The OCTN transporters, encoded by the *SLC22* genes, have been identified in the liver, kidney, and intestine [45]. OCTNs transport cationic substrates and carnitine [44]. As carnitine influx transporters, placental OCTNs may function to supply carnitine for fetal development and placental metabolism [16].

#### 3.2.4. Organic Anion Transporter (OAT) Family

Among the transporters in the OAT or *SLC22* family, OAT1 and OAT3 have been widely studied due to their importance in drug transport [45]. Observed in the liver, kidney, intestine, central nervous system, skeletal muscle, heart, lung, pancreas, and adrenal gland, OATs are anion exchangers that pair the uptake of an anionic substrate with the efflux of another anion [44]. Though OAT1-3 are predominantly expressed in excretory organs [16], OAT4 mediates the transport of anions [27] and the uptake of estrogen precursors [46] in the placenta.

#### 3.2.5. Concentrative Nucleoside Transporter (CNT) Family

CNTs are members of the *SLC28* gene family and aid in the transport of nucleotides, nucleosides, and nucleoside analogs [47]. CNTs contribute to the biosynthesis, absorption, metabolism, and elimination of nucleotides in the brain, intestine, liver, and kidney, respectively [48]. Though limited expression of CNTs occurs in the placenta [34], CNT1 has been speculated to supply pyrimidine nucleosides for placental development [16].

#### 3.2.6. Equilibrative Nucleoside Transporter (ENT) Family

ENTs are members of the *SLC29* gene family, facilitating the transport of nucleosides, nucleobases, and monoamines [47]. Like CNTs, ENTs have been identified in the brain, intestine, liver, and kidney [34,48]. While ENT expression has been observed in the placenta, it is unclear whether these transporters are functional [34].

#### 3.2.7. Multidrug and Toxin Extrusion (MATE) Family

The MATE transporters, encoded by the *SLC47* genes, efflux organic cations from cells, often in tandem with the influx activity of OCTs [44]. MATE1 is expressed in the liver, kidney, skeletal muscle, and adrenal gland. MATE2 has two additional protein variants, MATE2-B and MATE2-K, with MATE2-K specifically expressed in the kidney. Based upon the observed interaction between MATE1 and OCT3 in the rat placenta, coupled transport of organic cations is also predicted to occur in the human placenta [49].

### 3.3. Neonatal Fc Receptor (FcRn)

FcRn is an immunoglobulin G (IgG) receptor that is distributed in the liver, kidney, lung, and skin [50]. In the placenta, FcRn mediates the transplacental transport of IgG from the mother to the fetus (Figure 1) [36]. As suggested with in vitro experiments, the mechanism of IgG transport involves endocytosis of the IgG–FcRn complex into acidic endosomes, transcytosis of the complex to the opposite membrane, and pH-triggered dissociation of the complex upon membrane fusion [51,52]. In addition to facilitating the transplacental transfer of IgG, the FcRn system may also enable the transplacental transport of monoclonal antibodies, which may increase fetal drug exposure or necessitate therapeutic drug monitoring during pregnancy [53].

## 4. Phase II Enzymes

Phase II drug metabolism involves the conjugation of large moieties to a parent drug or drug metabolite, facilitating the biological inactivation of drug molecules and/or increasing their aqueous solubility for urinary or biliary excretion [1]. Conjugation reactions are catalyzed by phase II drug-metabolizing enzymes, as described in this section.

### 4.1. Methyltransferase (MT) Superfamily

MTs transfer a methyl group from a donor molecule to a substrate [54,55] and are located in cells of the liver, kidney, intestine, brain, and blood [56]. In the placenta, MTs may affect the homeostasis of the human chorionic gonadotropin (HCG) hormone [54] and may contribute to placental and embryonic development [55].

### 4.2. Glutathione S-Transferase (GST) Superfamily

Enzymes in the GST superfamily catalyze the conjugation of glutathione moieties to xenobiotic electrophiles or reactive oxygen species [57]. GSTs are distributed in the liver, kidney, intestine, brain, heart, lung, pancreas, and spleen [58]. Placental GSTs detoxify and bioactivate xenobiotics [57].

### 4.3. N-Acetyltransferase (NAT) Superfamily

NATs transfer an acetyl group to a nitrogen acceptor of primary arylamines and hydrazines [59]. The NAT superfamily includes N-acetyltransferase 1 (NAT1), which is expressed in the intestine, bladder, and breast, and N-acetyltransferase 2 (NAT2), which is expressed in the liver and intestine [60]. Both NAT1 and NAT2 are expressed in the placenta, with NAT1 providing a greater contribution to placental acetylation capacity [59].

### 4.4. Sulfotransferase (SULT) Superfamily

Enzymes in the SULT superfamily catalyze the sulfonation of endogenous substrates and xenobiotics [56]. SULTs have been identified in cells of the liver, kidney, intestine, brain, blood, lung, adrenal gland, breast, endometrium, and ovary, though tissue distribution varies between SULT subfamilies. Placental SULTs catalyze the biotransformation of estrogens to regulate intracellular steroid concentrations [10,61].

### 4.5. UDP-Glucuronosyltransferase (UGT) Superfamily

UGTs catalyze the addition of glucuronic acid to endogenous substrates and hydrophobic drug molecules, forming β-D-glucuronide metabolites [62]. Though mostly distributed in the liver, UGTs have also been observed in the gastrointestinal tract, kidney, brain, lung, pancreas, breast, and nasal epithelium. In the placenta, UGTs may participate in the metabolism of steroid substrates, thyroid hormones, and bile acids [63].

## 5. Materials and Methods

This review is guided by the 2020 Preferred Reporting Items for Systematic reviews and Meta-Analyses (PRISMA) statement [64], which was utilized to identify literature relevant to phase II drug metabolism and drug transport in pregnancy. The search strategy was conducted using PubMed^®^, a citation database for biomedical literature. Searches were completed during the period of 26 May 2023 to 10 July 2023.

Using the “AND” and “OR” functions in PubMed, the 51 search terms listed in Table 1 were employed to identify the initial dataset of manuscripts. The title of the manuscript contained the word “pregnancy”, “pregnant”, “placenta”, or “placental” and at least one of the remaining words listed in Table 1. To remove irrelevant manuscripts, titles that contained at least one of the 34 terms listed in Table 2 were excluded using the “NOT” function in PubMed. The filters “Humans” and “English” were further applied to remove animal data and non-English manuscripts. Manuscripts that were not automatically removed via the search strategy were manually removed; one reviewer independently screened the remaining abstracts for the words listed in Table 2, and the corresponding manuscripts were excluded from the dataset. Non-primary literature, such as literature reviews, were also excluded. The search strategy and ineligibility criteria are detailed in Figure 2, in which “Identification” details the development of an initial dataset of manuscripts, “Screening” encompasses the exclusion of irrelevant manuscripts, and “Inclusion” describes the evaluation of the final dataset of manuscripts.

One reviewer independently collected data from the final dataset of manuscripts. The manuscripts were evaluated for evidence regarding gestational changes in the expression or activity of drug transporters in the pregnant woman (i.e., maternal drug transporters) and in the placenta (i.e., placental drug transporters). In each gestational trimester, as well as postpartum (≥4 weeks postdelivery), transporter expression or activity was designated as increased, decreased, or unchanged compared to the previous gestational stage. Trimester 1 was compared to preconception or postpartum, assuming that baseline was achieved after delivery; trimester 2 was compared to trimester 1; trimester 3 was compared to trimester 2; and postpartum was compared to trimester 3. Gestational stages were defined according to recommendations by the United States Department of Health and Human Services: the first trimester was defined as gestational week 0–12, the second trimester as gestational week 13–28, and the third trimester as gestational week 29–40 [65]. Where data were available, the effect of hormones (e.g., 17β-estradiol and progesterone) and pregnancy complications on transporter expression or activity were also collected. The same parameters were evaluated for maternal and placental phase II enzymes.

## 6. Results

A total of 142 studies were included in this literature review (Figure 2). From these studies, 16 phase II drug-metabolizing enzymes and 38 drug transporters were identified as demonstrating evidence regarding the effect of gestational age, pregnancy complications, and/or hormones on maternal and/or placental protein expression or activity. 

Table 3 summarizes the effects of 17β-estradiol and progesterone on mRNA expression, protein expression, and/or activity of select phase II enzymes and drug transporters during pregnancy. These data have been utilized to reveal potential regulatory mechanisms of metabolism and transport, as well as to supplement the collected evidence for gestational changes in protein expression or activity.

Table 4 qualitatively illustrates the gestational changes in the expression or activity of maternal and placental phase II enzymes, and Table 5 qualitatively illustrates the gestational changes in the expression or activity of maternal and placental drug transporters. These changes are detailed across the first, second, and third trimesters of pregnancy, as well as postpartum. For several phase II enzymes and drug transporters, evidence of protein expression or activity was not detailed in the manuscripts of the dataset; these enzymes and transporters have not been included in Table 4 or Table 5, but additional information can be found in Appendix A.

Appendix A includes the complete list of phase II enzymes (n = 16) and drug transporters (n = 38) for which at least one relevant manuscript was identified. Protein function and pharmaceutical substrates that were studied in relation to pregnancy have been detailed. Appendix A describes quantitative evidence regarding the effect of gestational age and hormones on the expression or activity of maternal phase II enzymes, expands upon the physiological location of the enzymes, and provides references for the cited information. Appendix A focus on placental phase II enzymes and maternal drug transporters, respectively, detailing the same parameters as Appendix A. Appendix A describes information regarding gestational age, hormones, physiological location, and transport directionality as they relate to placental drug transporters. Data involving the effect of pregnancy complications on enzyme or transporter expression and activity were collected but not included in the analysis; the complete results can be found in Appendix A.

The tables in Appendix A summarize findings for each phase II enzyme (Appendix A) and drug transporter (Appendix A) studied and comment on the implications of the findings in the context of related research. 

The final dataset of 142 journal articles is cited in Appendix A. All citations in the Appendix A correspond with the numbered references in Appendix A.

## 7. Discussion

### 7.1. Effects of 17β-Estradiol and Progesterone on Drug Metabolism and Transport

Physiological concentrations of 17β-estradiol and progesterone increase throughout pregnancy until term [96], achieving peak concentrations of 50–100 nM of 17β-estradiol and 10–500 nM of progesterone [21,67]. Considering these gestational increases, in vitro observations of the hormonal effects for UGT1A4, BSEP, MRP3, ENT1, and MRP1 (Table 3) have been utilized to postulate the gestational changes in enzyme or transporter expression and activity (Table 4 and Table 5), in the absence of other available evidence to describe the changes.

Upregulation of UGT1A4 mRNA expression and lamotrigine glucuronidation (Table 3) was observed in HepG2 cells exposed to the levels of 17β-estradiol seen in pregnancy [66]. The increases in UGT1A4 expression and metabolic activity, as well as clinically observed higher lamotrigine oral clearance, support the contention that hepatic UGT1A4 activity may be increased during pregnancy (Table 4), especially considering that the secretion of estrogens increases until term.

17β-Estradiol decreased BSEP mRNA expression and protein expression (Table 3) in primary hepatocytes, which was observed to be a concentration-dependent change [68]. Increasing levels of 17β-estradiol until peak concentrations at term may suggest a decrease in hepatic BSEP mRNA expression and protein expression across gestation (Table 5). Additional PK data are required to confirm these assumptions.

MRP3 mRNA expression and protein expression were upregulated (Table 3) in hepatic LO2 cells that were treated with 500 nM of 17β-estradiol [69]. It should be noted that this concentration is higher than that which is physiologically attainable during pregnancy; since lower concentrations were not studied, it is unclear whether in vivo upregulation of hepatic MRP3 occurs across the three gestational trimesters (Table 5).

17β-Estradiol and progesterone downregulated ENT1 transport activity (Table 3) in transfected HEK293 cells, though the concentrations that attained significant reductions in activity (i.e., 1–100 μM of 17β-estradiol and 10–100 μM of progesterone) were higher than clinically attainable concentrations during pregnancy [73]. It is uncertain whether the decreased in vitro observations are representative of clinical findings for hepatic ENT1 activity across gestation (Table 5).

In placental trophoblasts treated with 100 nM of progesterone, an attainable concentration during gestation, placental MRP1 mRNA expression was upregulated, but protein expression was not significantly changed (Table 3) [21]. Due to inconsistencies in the effect of progesterone on MRP1 mRNA expression and protein expression, it has been assumed that the change in MRP1 protein expression, rather than mRNA expression, may be more predictive of the change in MRP1 activity. As MRP1 protein expression was not evaluated at higher concentrations of progesterone, further evidence is necessary to confirm a lack of change across gestational stages (Table 5).

Additional experiments conducted with physiologically attainable hormonal concentrations and data collected across gestational trimesters can verify the postulated gestational changes in the expression or activity of UGT1A4, BSEP, MRP3, ENT1, and MRP1.

### 7.2. Effects of Gestational Age on Phase II Enzyme Expression or Activity

#### 7.2.1. Maternal Phase II Enzyme Expression or Activity

In the following section, clinical observations of maternal plasma concentrations of phase II enzymes or their probe substrates have been utilized as evidence to support the analysis of maternal phase II enzyme expression or activity.

While the evidence agrees that there is no significant change in maternal glutathione S-transferase α (GSTA) enzyme expression from preconception through the third trimester of pregnancy [74,75], Zusterzeel et al. also observed an approximate 2-fold increase in plasma GSTA concentration from gestational week 30 to postpartum week 6 (Table 4) [74]. No mechanistic hypothesis was provided for this increase in enzyme expression, and further research may be required to verify these findings.

PK data collected following administration of hydralazine [76] and caffeine [77] as probe substrates of NAT2 suggest that maternal NAT2 activity decreases in the first trimester, returns to baseline in the second trimester, and remains at baseline in late pregnancy and postpartum (Table 4). While the 13% reduction in first trimester NAT2 activity demonstrated statistical significance when compared to NAT2 activity at 4–6 weeks postpartum [77], additional research is necessary to understand the clinical implications of these gestational changes to drug safety and efficacy.

The search strategy utilized in this literature review was unable to identify studies describing individual changes in activity for UGT1A1, UGT1A3, and UGT2B7, though a study involving buprenorphine metabolism was retrieved. Considering buprenorphine metabolism by UGT1A1, UGT1A3, and UGT2B7, combined maternal enzyme activity appears to increase from the first trimester through the third trimester, then returns to baseline postdelivery; these conclusions are based upon observed changes in the area under the curve (AUC) ratio of buprenorphine glucuronide metabolite to buprenorphine parent drug across gestational trimesters [78]. Due to the complex contribution of UGT1A1, UGT1A3, and UGT2B7 to buprenorphine metabolism, it is difficult to conclude which enzyme(s) are responsible for the changes in glucuronidation and, thus, which enzyme(s) exhibit altered activity during gestation. Consequently, UGT1A1, UGT1A3, and UGT2B7 have been grouped together in this analysis, and the reported changes in activity for the individual enzymes represent their combined activity (Table 4). Further investigation should be conducted to elucidate the individual changes in the activity of UGT1A3 and UGT2B7, which cannot be determined by buprenorphine metabolism alone. However, while no evidence supporting individual changes in UGT1A1 activity was identified via the search strategy in this review, pregnancy PK data have been published for some probe drugs of UGT1A1 (e.g., raltegravir). Watts et al. reported an approximate 50% reduction in median raltegravir AUC values in women during the second and third trimesters of pregnancy compared to the same women at 6-12 weeks postpartum [97]. The observed gestational decrease in raltegravir systemic exposure suggests an increase in UGT1A1 activity during pregnancy, a change that is currently incorporated into PBPK platforms [98].

Because phase II enzymes are critical components in the elimination of many drugs and their metabolites, changes in their activity during pregnancy present consequences to drug safety and efficacy for the pregnant woman. A phase II enzyme that exhibits a gestational decrease in activity will demonstrate decreased conversion of a substrate to its easily eliminated form [1]. This decrease in hepatic clearance may result in increased systemic exposure of the parent drug, leading to adverse effects and potential maternal toxicity if the dose is not adequately adjusted [76]. Conversely, a gestational increase in phase II enzyme activity may lead to increased hepatic clearance, decreased drug exposure, subtherapeutic drug concentrations, and impaired disease control [4]. For drugs that are metabolized by enzymes with increased activity, monitored dose escalation may be required during pregnancy, followed by monitored de-escalation postpartum [8]. Furthermore, it is unclear whether gestational stage-specific changes in drug-metabolizing enzyme activity may necessitate multiple dose adjustments throughout pregnancy [5]. Incorporating changes in maternal phase II enzyme activity into PBPK models will improve predictions of PK changes during pregnancy and enable well-informed dose adjustments.

#### 7.2.2. Placental Phase II Enzyme Expression or Activity

As suggested with analysis of placental tissue samples, placental protein carboxyl-O-methyltransferase (CMT) activity increases throughout the first trimester, achieves a peak at the beginning of the second trimester, then decreases until term (Table 4) [54]. Apart from CMT, there is limited data available to support changes in the expression or activity of placental phase II enzymes across gestational trimesters; for placental GSTA, glutathione S-transferase μ (GST-μ), glutathione S-transferase π (GST-π), and NAT1, evidence was only found for one of the three trimesters, and no postpartum evidence was found for any phase II enzyme studied (Table 4).

Though it has been suggested that placental phase II enzymes may marginally contribute to maternal drug metabolism and PK [9], placental enzyme activity has greater toxicologic implications for the fetus [17]. For instance, UGTs are predicted to contribute to the fetoprotective metabolism of foreign compounds during embryogenesis and organogenesis [17]. Compared to placental drug transport, however, placental drug metabolism has a relatively minor contribution to fetal drug exposure and is often not included in current pregnancy PBPK models [99].

### 7.3. Effects of Gestational Age on Drug Transporter Expression or Activity

#### 7.3.1. Maternal Drug Transporter Expression or Activity

In the following section, clinical PK data for maternal drug transporters are detailed to support the analysis of maternal drug transport expression or activity.

P-gp appears to exhibit tissue-specific changes in activity from the third trimester of pregnancy to postpartum (Table 5). In a digoxin probe study, renal P-gp activity was observed to decrease; the mean renal secretion clearance of digoxin was 120% higher for pregnant women in gestational week 28–32 compared to postpartum week 6–10 [79]. In a fexofenadine probe study, no significant change in intestinal P-gp activity was observed; ratios comparing third trimester to postpartum fexofenadine Cmax and AUC values were within the equivalence window of 0.8–1.25 [80].

While additional data are necessary to confirm renal OAT1, OAT2, and OAT3 activity during pregnancy, the transporters appear to behave differently across gestational trimesters [92]. Peak transporter activity is predicted to occur in the second trimester, third trimester, and first trimester for OAT1, OAT2, and OAT3, respectively, before subsequent declines to baseline (Table 5). 

No studies were found regarding individual changes in activity during pregnancy for BCRP, OATP1B1, and OATP1B3. As suggested with rosuvastatin PK data, the collective hepatic activity of BCRP, OATP1B1, and/or OATP1B3 is decreased in the third trimester, returning to baseline by postpartum; these conclusions are based upon the higher mean rosuvastatin AUC values observed in pregnant women during the third trimester compared to AUC values observed at term and postpartum [80]. The complex contribution of BCRP, OATP1B1, and OATP1B3 to rosuvastatin transport renders it difficult to conclude which transporter(s) are responsible for increased drug exposure during the third trimester. OATP1B1 and OATP1B3 have been observed to contribute to more than 50% of total hepatic rosuvastatin uptake [100]; OATP1B3 is estimated to contribute to 16–34% of uptake, leaving 66–84% assumed to be attributable to OATP1B1 and/or other transporters [101]. In OATP1B1/BCRP double-transfected MDCKII cells, basal-to-apical efflux clearance was 1.7-fold higher than apical-to-basal influx activity [101], suggesting that the efflux activity of BCRP may have a greater contribution to rosuvastatin transport than the influx activity of OATP1B1. Because the fractional contribution of BCRP, OATP1B1, and OATP1B3 to rosuvastatin transport cannot be definitively deduced, these transporters have been grouped together for this analysis, and the reported changes in activity for the individual transporters represent their combined activity (Table 5). Additional PK studies that isolate the individual transporter activity of BCRP, OATP1B1, and OATP1B3 may aid in understanding the gestational changes of these transporters.

Limited data are available regarding the individual contributions of OCT2, MATE1, and MATE2-K to the vectorial transport of substrates [102]. Considering the transport of metformin and N1-methylnicotinamide by renal OCT2, MATE1, and MATE2-K, as well as the changes in the secretion clearance of the two probe drugs across gestation, combined transporter activity is suggested to increase from the first trimester to the second trimester and decrease from the third trimester to postpartum [90]. Activity differences between the second and third trimesters are inconclusive and appear to be substrate-specific: a decrease in mean secretion clearance was observed for metformin, indicating decreased renal transporter activity, while an increase in mean secretion clearance was observed for N1-methylnicotinamide, suggesting increased renal transporter activity [90]. It is difficult to conclude which transporter(s) are responsible for the changes in probe secretion clearance throughout pregnancy, especially considering the coupled transport mechanism of renal OCT2, MATE1, and MATE2-K; renal OCT2 is believed to facilitate the basolateral uptake of cationic substrates, such as metformin, while MATE1 and MATE2-K facilitate the coupled luminal secretion [103]. Further complicating interpretation of the metformin PK data is the observation that metformin is also transported by OCT1, OCT3, and ENT4 [104]. Due to limited evidence regarding the individual contributions of OCT2, MATE1, and MATE2-K to drug transport, the gestational changes in combined enzyme activity have been reported (Table 5) until more conclusive evidence can suggest unique changes in individual enzyme activity. 

Gestational changes in drug transporter activity have similar implications to drug safety and efficacy as gestational changes in phase II enzyme activity. A renal drug transporter that exhibits a gestational decrease in apparent activity may decrease renal clearance, increase drug exposure, and elevate the risk of drug toxicity [5]. Conversely, a gestational increase in renal drug transporter activity may increase renal clearance, decrease drug exposure, and limit dose efficacy. To maintain plasma drug concentrations within the therapeutic window throughout drug therapy, dose and/or dose frequency may require modification [8]. In current pregnancy PBPK models for renally cleared drugs, altered plasma drug concentrations may be difficult to predict, with mispredictions attributed to the omission of most drug transporter changes in the models [6]. Although gestational changes in OCT2 activity are known and currently incorporated into PBPK platforms, the gestational changes of other transporters have yet to be included. Understanding the changes in maternal drug transporter activity will allow for optimized dose modifications and PBPK model predictions in the pregnant population.

#### 7.3.2. Placental Drug Transporter Expression or Activity

Placental tissue obtained following delivery or termination of pregnancy, as well as in vitro cell models, have been utilized to study gestational changes in drug transporter activity, and application of the models is detailed in this section. Most of the evidence supporting placental transporter expression and activity is biological (i.e., mRNA expression, protein expression, and intracellular accumulation of probe molecules). mRNA expression and protein expression are assumed to translate to transporter activity unless evidenced otherwise.

Overall, placental P-gp activity demonstrates peak activity in the first trimester, followed by a decrease through the third trimester (Table 5) [81,82,83,84]. Inconclusive evidence is available regarding changes in transporter activity from the first to the second trimester; while Gil et al. observed a progressive trend towards decreasing P-gp protein expression during the second trimester [81], Sun et al. observed no significant difference in mRNA or protein expression between the first and second trimester [82]. With respect to hormonal regulation via 17β-estradiol, placental P-gp mRNA expression and protein expression increased in both placental JAR cells and isolated cytotrophoblasts, and the upregulation of P-gp expression translated to an upregulation of P-gp efflux activity in isolated cytotrophoblasts (Table 3) [21,67]. In response to progesterone, placental JAR cells also demonstrated similar behavior as isolated cytotrophoblasts: in both cell types, P-gp mRNA expression was not observed to change, while P-gp protein expression was observed to increase (Table 3). However, in placental BeWo cells, P-gp efflux activity appeared to be inhibited following exposure to progesterone (Table 3) [18]. Because placental BeWo cells are morphologically related to placental cytotrophoblasts and are predicted to model in vivo conditions [18], the data involving P-gp efflux activity in placental BeWo cells may reasonably be compared to the expression data observed in placental JAR cells or isolated cytotrophoblasts. The inconsistency between the upregulation of protein expression and the downregulation of efflux activity may suggest a complex interaction between progesterone and the placental P-gp transporter. Despite uncertainties in the regulatory mechanism of P-gp expression, the general trend towards decreasing placental P-gp activity across pregnancy may inform PBPK model predictions of fetoplacental drug exposure for P-gp substrates.

Placental MDR3 expression across each gestational trimester cannot be deduced, though evidence is available to describe a general trend of expressional change from the first trimester through the third trimester. However, this evidence is contradictory, revealing both increased gene expression [85] and decreased gene expression [40] between first and third trimester placental samples (Table 5). The effect of 17β-estradiol on placental MDR3 mRNA and protein expression may aid in understanding the inconsistencies. Exposure of placental trophoblasts to 100 nM 17β-estradiol, an attainable concentration during gestation, increased MDR3 mRNA expression and protein expression by more than 60% compared to untreated cells (Table 3) [21]. Since an increase in physiological 17β-estradiol concentrations is expected until term, the evidence obtained from placental trophoblasts supports an increase in placental MDR3 expression throughout pregnancy. Continued investigation of placental MDR3 is required to conclude the expressional pattern across gestation and to evaluate whether changes in transporter expression translate to changes in transporter activity.

Gestational changes in placental MRP2 expression cannot be determined due to conflicting evidence (Table 5). In one study conducted by Meyer zu Schwabedissen et al., placental MRP2 mRNA expression was demonstrated to increase from the second trimester to term [22], but in a separate study, Imperio et al. observed a decrease in mRNA expression from the first trimester to term [40]. While these data may not be mutually exclusive, additional research of placental MRP2 will solidify current knowledge of transporter expression and activity.

Compared to MRP5 mRNA expression in placentas at gestational week 23–32, mRNA expression was significantly lower in placentas at gestational week 32–37 and 6-fold lower in placentas at term (Table 5) [25]. Though evidence is lacking for placental MRP5 expression in early pregnancy, the available evidence for decreased expression in late pregnancy may provide a starting point for the incorporation of MRP5 gestational changes in PBPK models.

Gestational changes in placental BCRP expression remain uncertain (Table 5). Petrovic et al. reported a decrease in BCRP mRNA expression in placental samples across the third trimester [86]. Yeboah et al. observed no significant change in placental BCRP mRNA expression throughout gestation but an increase in placental BCRP protein expression at the end of gestation [87]. Although Scott et al. observed a similar pattern in BCRP mRNA expression across gestation, lower BCRP protein expression was observed in term pregnancy samples compared to preterm pregnancy samples [84]. Other studies also support the decreasing trend in BCRP protein expression by term, when comparing term placental samples to third trimester samples [86] or first trimester samples [88]. In addition to inconsistencies in the evidence of placental BCRP mRNA and protein expression, evidence of placental BCRP hormonal regulation also describes opposing trends. Downregulation of BCRP protein expression and efflux activity in response to 17β-estradiol (Table 3) was detected in BeWo cells [70], findings that were supported by the decrease in BCRP mRNA expression observed in placental explants [21]. Progesterone exhibited the opposite effect on BCRP in placental BeWo cells: significant increases in mRNA expression [71], protein expression [70], and efflux activity were observed (Table 3). Taken together, these data indicate limited understanding of the effect of gestational age on BCRP expression and activity. As BCRP is an important transporter for various drug substrates, incorporation of BCRP into pregnancy PBPK models could improve the accuracy of PK predictions for many drugs. However, confidence in these models can only be achieved with adequate knowledge of BCRP expression and activity across pregnancy.

Contradictory evidence regarding the gestational changes in placental OATP2B1 expression does not allow for definitive determination of transporter changes during pregnancy (Table 5). Placental OATP2B1 protein abundance was demonstrated to decrease from the first trimester to the second trimester [83], and mRNA expression was observed to decrease throughout the third trimester [86]. Conversely, Nabekura et al. presented contradictory findings, suggesting a 2.5-fold increase in OATP2B1 mRNA expression in term placental samples compared to first trimester placental samples [89]. Further investigation of placental OATP2B1 is required to conclude the patterns in transporter expression and activity during pregnancy.

A consistent increase in placental OCT3 protein expression from the first trimester to term has been suggested [83,91]. Placental OCT3 mRNA expression, on the other hand, was observed to be significantly decreased in term placentas compared to first trimester placentas [49]. Inconsistencies between protein expression and mRNA expression (Table 5) suggest that changes at the molecular level may not translate to changes at the protein level or activity level. Changes in protein expression may be a more predictive indicator of changes in transporter activity, and additional research will aid in confirming or disproving an increase in placental OCT3 activity across gestation.

Decreased placental OCTN2 mRNA expression, though not considered to be statistically meaningful, was observed in third trimester placental samples compared to term placental samples [31]. The lack of clear results for OCTN2 mRNA expression demonstrates insufficient evidence to support gestational changes in transporter expression from the second trimester to the third trimester; from the available data, it can only be concluded that placental OCTN2 mRNA expression does not appear to change throughout the third trimester (Table 5). Considering the importance of OCTN2 to placental carnitine transport, the lack of change in late pregnancy may indicate that changes occurred earlier in pregnancy, though specific changes could not be elucidated in this literature review.

Placental OAT4 activity appears to increase from the second to the third trimester (Table 5), as suggested by a 1.6-fold increase in protein abundance at term compared to the second trimester [83]. With respect to the hormonal regulation of OAT4, transporter activity in OAT4-expressing placental BeWo cells was unaffected by 17β-estradiol (Table 3) but was downregulated by progesterone [72]. It should be noted that the significant downregulation of OAT4 activity was only observed at a high concentration of progesterone (10 μM); though physiologically attainable progesterone concentrations were studied, no change in OAT4 activity was observed, suggesting that placental OAT4 activity may not exhibit gestational changes (Table 3). Understanding the extent of the contribution of hormonal regulation to OAT4 activity in vivo may assist in the confident conclusion of gestational changes for OAT4.

Increases in fetal IgG concentration and fetal-to-maternal IgG ratio with increasing gestational age suggest increased placental FcRn transport capacity throughout pregnancy [95]. To facilitate the increase in placental FcRn transport capacity, the expression or activity of the FcRn receptor likely also increases with increasing gestational age (Table 5). Understanding the gestational changes in FcRn transporter activity and, therefore, the efficiency of transplacental drug transport may optimize PBPK models for monoclonal antibodies, especially as it relates to drug efficacy for the pregnant woman and drug safety to the fetus.

Many of the placental transporters for which data were found demonstrate a general trend of increased (e.g., MRP3, CNT2, and CNT3), decreased (e.g., BSEP, OATP1A2, OATP3A1, and OCT2), or unchanged (e.g., ENT1 and ENT2) transporter expression or activity from the first trimester to term (Table 5). These gestational changes can be utilized as a preliminary foundation to inform PBPK model optimization until more data are uncovered. Other placental transporters, as described in this section, lack robust data to formulate a conclusive pattern of gestational expression or activity, and more studies should be conducted with these transporters in each gestational trimester to ensure confident incorporation of their activity into PBPK models. 

Increased knowledge of placental transporter activity will inform understanding of PK changes in the pregnant woman, as well as potential adverse event risk for the fetus [10]. Drugs can be transferred across the placental syncytiotrophoblast in the maternal-to-fetal or the fetal-to-maternal direction, which increases or decreases, respectively, drug exposure in the fetoplacental compartment [17]. Thus, understanding transplacental drug distribution will support the robustness of PBPK model predictions, including predictions of maternal PK changes and fetal drug exposure.

### 7.4. Limitations

The screening of manuscripts and the collection of data were not confirmed by a second individual. Because the search was solely conducted using PubMed, there is also the risk that the search criteria did not encapsulate all relevant literature about gestational changes in phase II enzyme and drug transporter activity. To ensure completeness, the search term “pharmacokinetics” was checked in addition to those listed in Table 1, but the search returned limited results regarding transporter-associated PK.

PK data were available for the maternal phase II enzymes and drug transporters; the summarized PK data can be utilized to aid pregnancy PBPK modeling when no clinical data exist to guide dosing. However, in the absence of available PK data for placental phase II enzymes and drug transporters, in vitro observations (e.g., changes in mRNA expression, protein expression, and/or protein activity observed in placental tissue explants or placental cell lineages) have been utilized to formulate conclusions of gestational changes in placental enzyme and transporter expression or activity. Changes in mRNA expression or protein expression were also assumed to translate to changes in enzyme or transporter activity unless demonstrated otherwise. The assumptions made based upon in vitro observations may require verification with additional evidence before confident application into PBPK models. Specific assumptions are identified in the Discussion.

## 8. Conclusions

Sparse and conflicting evidence exists regarding the localization, expression, and regulation of phase II drug-metabolizing enzymes and drug transporters across gestational stages. To resolve the uncertainties and assumptions in current knowledge, additional PK data and clinical pharmacology research are required to understand drug metabolism and transport in the pregnant woman and in the placenta. Although this comprehensive review of phase II enzymes and drug transporters exposes gaps in knowledge, the available information of enzyme and transporter changes may aid in optimizing pregnancy PBPK models to improve the accuracy of model predictions, confirm the need for dose adjustments, and optimize the safety of the developing fetus.

## Figures and Tables

**Figure 1 pharmaceutics-15-02624-f001:**
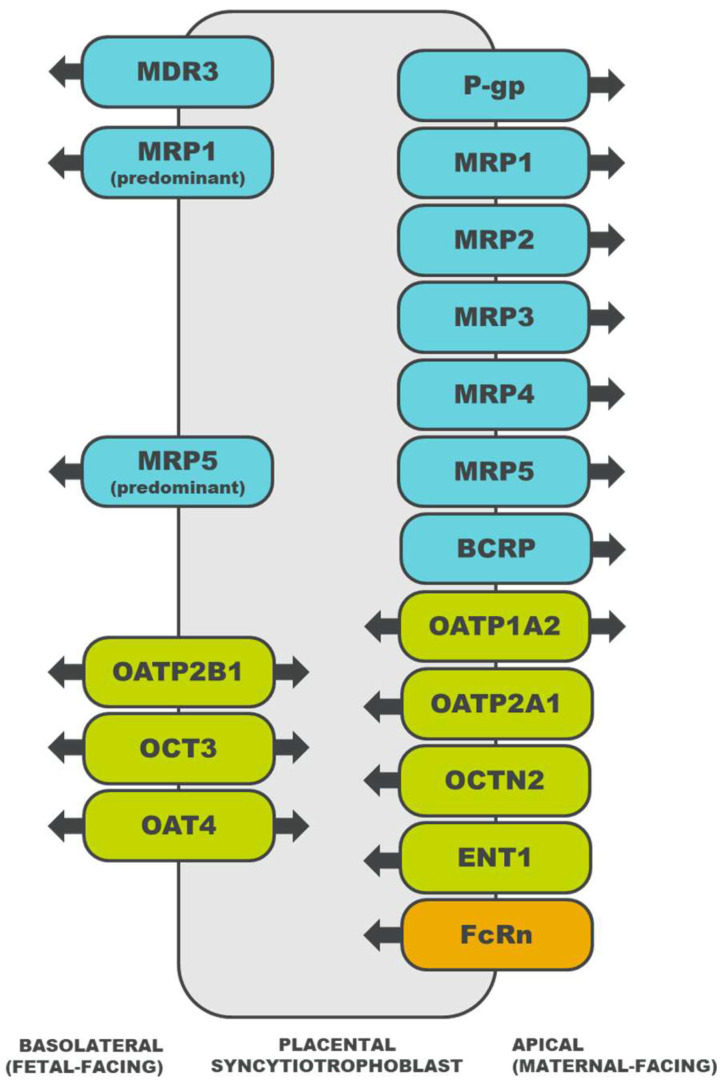
Localization of drug transporters in the placental syncytiotrophoblast and the directionality of their transport [18,19,20,21,22,23,24,25,26,27,28,29,30,31,32,33,34,35,36]. ATP-binding cassette (ABC) transporters are depicted in blue, solute carrier (SLC) transporters in green, and an immunoglobulin transporter in orange. MDR, multidrug resistance protein; P-gp, P-glycoprotein; MRP, multidrug resistance associated protein; BCRP, breast cancer resistance protein; OATP, organic anion transporting polypeptide; OCT, organic cation transporter; OCTN, organic cation/carnitine transporter; OAT, organic anion transporter; ENT, equilibrative nucleoside transporter; and FcRn, neonatal Fc receptor.

**Figure 2 pharmaceutics-15-02624-f002:**
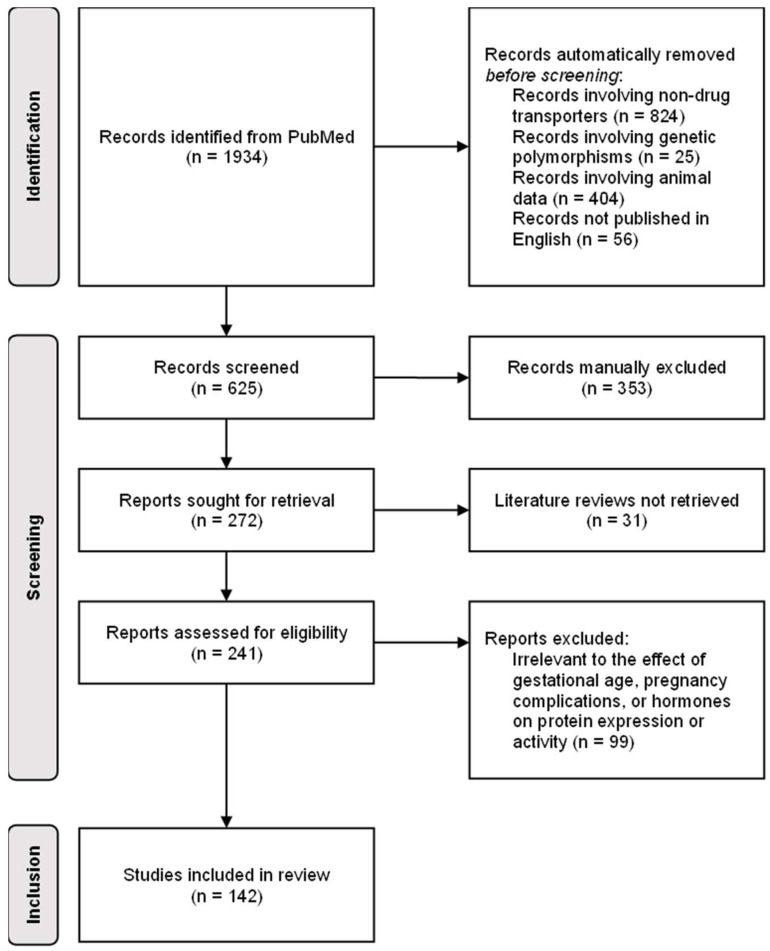
Preferred Reporting Items for Systematic reviews and Meta-Analyses (PRISMA)-guided flow diagram for identifying reports to review.

**Table 1 pharmaceutics-15-02624-t001:** PubMed^®^ search terms that were utilized to identify the initial dataset of manuscripts.

pregnancy	MRP2	OCT3	glucuronosyltransferase
pregnant	MRP3	OCTN1	UGT
placenta	MRP4	OCTN2	UGT1A1
placental	MRP5	OAT1	UGT1A4
transporter(s)	MRP6	OAT2	UGT1A9
transport	breast cancer resistance protein	OAT3	UGT2B4
P-glycoprotein	BCRP	CNT1	UGT2B7
P-gp	OATP1B1	ENT1	UGT2B15
MDR1	OATP1B3	ENT2	sulfotransferase
MDR3	OATP2B1	MATE1	N-acetyltransferase
bile salt export pump	OATP4A1	MATE2	S-transferase
BSEP	OCT1	PEPT1	methyltransferase
MRP1	OCT2	PEPT2	

**Table 2 pharmaceutics-15-02624-t002:** PubMed^®^ terms that were automatically excluded prior to screening.

nutrient	folate	sodium
mineral	metal	sulfate
amino acid(s)	cadmium	sulphate
taurine	calcium	zinc
creatine	chloride	choline
glucose	chromium	norepinephrine
fatty acid(s)	copper	serotonin
lipid	iron	monoamine
cholesterol	magnesium	thyroid
vitamin	phosphate	polymorphism(s)
thiamine	potassium	
riboflavin	selenium	

**Table 3 pharmaceutics-15-02624-t003:** The effects of 17β-estradiol and progesterone on mRNA expression, protein expression, and/or activity of phase II drug-metabolizing enzymes and drug transporters during pregnancy [18,21,24,66,67,68,69,70,71,72,73].

		Effect of 17β-Estradiol			Effect of Progesterone			
Enzyme or Transporter	Location	mRNA	Protein	Activity	mRNA	Protein	Activity	Citations
Phase II Enzyme								
UGT1A4	liver	↑	-	↑	-	-	-	[66]
Drug Transporter								
P-gp	placenta	↑	↑	↑	↔	↑	↓	[18,21,67]
MDR3	placenta	↑	↑	-	-	-	-	[21]
BSEP	liver	↓	↓	-	-	-	-	[68]
MRP1	placenta	-	-	substrate	↑	↔	-	[21,24]
MRP3	liver	↑?	↑?	-	-	-	-	[69]
BCRP	placenta	↓	↓	↓	↑	↑	↑	[21,70,71]
OAT4	placenta	-	-	↔?	-	-	↔?	[72]
ENT1	liver	-	-	↓?	-	-	↓?	[73]

UGT, UDP-glucuronosyltransferase; P-gp, P-glycoprotein; MDR, multidrug resistance protein; BSEP, bile salt export pump; MRP, multidrug resistance associated protein; BCRP, breast cancer resistance protein; OAT, organic anion transporter; ENT, equilibrative nucleoside transporter; ↑, upregulation; ↓, downregulation; ↔, no change; -, no or insufficient evidence found to make any assumptions; and ?, assumptions that require verification with additional evidence.

**Table 4 pharmaceutics-15-02624-t004:** Gestational changes in the expression or activity of phase II drug-metabolizing enzymes in the pregnant woman and in the placenta [54,57,59,66,74,75,76,77,78]. Changes in each gestational stage were compared to the previous stage, as described in Section 5. No data were available regarding postpartum changes in placental phase II enzymes.

	Expression or Activity in the Pregnant Woman				Expression or Activity in the Placenta			
Enzyme	Trimester 1	Trimester 2	Trimester 3	Postpartum	Trimester 1	Trimester 2	Trimester 3	Citations
CMT	-	-	-	-	↑	peak activity	↓	[54]
GSTA	↔	↔	↔	↑	↓	-	-	[57,74,75]
GST-μ	-	-	-	-	↔	-	-	[57]
GST-π	-	-	-	-	↔	-	-	[57]
NAT1	-	-	-	-	-	-	↑	[59]
NAT2	↓	↑ to baseline	↔	↔	-	-	-	[76,77]
UGT1A1 *	↑	↑	peak activity	↓ to baseline	-	-	-	[78]
UGT1A3 *	↑	↑	peak activity	↓ to baseline	-	-	-	[78]
UGT1A4	↑?	↑?	↑?	-	-	-	-	[66]
UGT2B7 *	↑	↑	peak activity	↓ to baseline	-	-	-	[78]

trimester 1, gestational week 0–12; trimester 2, gestational week 13–28; trimester 3, gestational week 29–40; postpartum, ≥ 4 weeks postdelivery; CMT, protein carboxyl-O-methyltransferase; GSTA, glutathione S-transferase α; GST-μ, glutathione S-transferase μ; GST-π, glutathione S-transferase π; NAT, N-acetyltransferase; ↑, increase; ↓, decrease; ↔, no change; -, no or insufficient evidence found to make any assumptions; and ?, assumptions that require verification with additional evidence. *, Maternal UGT1A1, UGT1A3, and UGT2B7 activity have been inferred from the combined evidence of buprenorphine metabolism. See Section 7 for further details.

**Table 5 pharmaceutics-15-02624-t005:** Gestational changes in the expression or activity of drug transporters in the pregnant woman and in the placenta [21,22,25,31,40,49,68,69,79,80,81,82,83,84,85,86,87,88,89,90,91,92,93,94,95]. Unless otherwise stated, changes in each gestational stage were compared to the previous stage, as described in Section 5. No data were available regarding postpartum changes in placental transporters. For cases in which conflicting evidence was found, multiple symbols have been included to reflect the inconsistencies.

	Expression or Activity in the Pregnant Woman				Expression or Activity in the Placenta			
Transporter	Trimester 1 (T1)	Trimester 2 (T2)	Trimester 3 (T3)	Postpartum	Trimester 1 (T1)	Trimester 2 (T2)	Trimester 3 (T3)	Citations
P-gp	-	-	-	↓ to baseline (renal) ↔ (intestinal)	peak activity	↓ ↔	↓	[79,80,81,82,83,84]
MDR3	-	-	-	-	-	-	↓ ↑ from T1	[40,85]
BSEP	↓?	↓?	↓?	-	-	-	↓ from T1	[40,68,85]
MRP1	-	-	-	-	↔?	↔?	↔?	[21]
MRP2	-	-	-	-	-	↑	↓ ↑	[22,40]
MRP3	↑?	↑?	↑?	-	-	-	↑ from T1	[40,69]
MRP5	-	-	-	-	-	↓ during T2	↓	[25]
BCRP *	-	-	↓	↑ to baseline	↔	↔	↓ ↑	[80,84,86,87,88]
OATP1A2	-	-	-	-	-	-	↓ from T1	[85]
OATP1B1 *	-	-	↓	↑ to baseline	-	-	-	[80]
OATP1B3 *	-	-	↓	↑ to baseline	-	-	-	[80]
OATP2B1	-	-	-	-	-	↓	↓ ↑	[83,86,89]
OATP3A1	-	-	-	-	-	-	↓ from T1	[85]
OCT2 ^	-	↑	↓ ↑	↓	-	-	↓ from T1	[49,90]
OCT3	-	-	-	-	-	↑	↓ ↑ from T1	[49,83,91]
OCTN2	-	-	-	-	-	-	↔ during T3	[31]
OAT1	-	peak activity	↓	↓ to baseline	-	-	-	[92]
OAT2	-	-	peak activity	↓ to baseline	-	-	-	[92]
OAT3	peak activity	↓	↓	↓ to baseline	-	-	-	[92]
OAT4	-	-	-	-	-	-	↑	[83]
CNT2	-	-	-	-	-	-	↑ from T1	[93]
CNT3	-	-	-	-	-	-	↑ from T1	[93]
ENT1	↓?	↓?	↓?	-	-	-	↔ from T1	[94]
ENT2	-	-	-	-	-	-	↔ from T1	[94]
MATE1 ^	-	↑	↓ ↑	↓	-	-	-	[90]
MATE2 ^	-	↑	↓ ↑	↓	-	-	-	[90]
FcRn	-	-	-	-	-	↑?	↑?	[95]

trimester 1, gestational week 0–12; trimester 2, gestational week 13–28; trimester 3, gestational week 29–40; postpartum, ≥ 4 weeks postdelivery; OATP, organic anion transporting polypeptide; OCT, organic cation transporter; OCTN, organic cation/carnitine transporter; CNT, concentrative nucleoside transporter; MATE, multidrug and toxin extrusion; FcRn, neonatal Fc receptor; ↑, increase; ↓, decrease; ↔, no change; -, no or insufficient evidence found to make any assumptions; and ?, assumptions that require verification with additional evidence. *, Maternal BCRP, OATP1B1, and OATP1B3 activity have been inferred from the combined evidence of rosuvastatin metabolism. See Discussion for further details. ^, Maternal OCT2, MATE1, and MATE2 activity have been inferred from the combined evidence of metformin and N1-methylnicotinamide metabolism. See Discussion for further details.

## Data Availability

The data presented in this review are available in Appendix A.

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
