# Peer review of "A Literature Review of Changes in Phase II Drug-Metabolizing Enzyme and Drug Transporter Expression during Pregnancy"

_pharmaceutics, 2023, doi:10.3390/pharmaceutics15112624_

Round 1
Reviewer 1 Report
Comments and Suggestions for Authors
The authors present a systematic review that summarizes current knowledge and gaps considering abundance of transporters and phase II drug metabolizing enzymes in pregnant women. Drug administration during pregnancy is a major challenge to avoid any potential harms to the growing fetus as well as to sustain therapeutic levels for the woman. Hence, the work is interesting considering the need to summarize the available evidence towards also to build validated pharmacometric approaches and PBPK models.
A major issue of the manuscript is that the authors state it is a systematic review. Although MEDLINE has the majority of available literature it is generally adviced that SRs should search more than one database i.e., Embase, MEDLINE, Web of Science, and/or Google Scholar as a minimum requirement for adequate and efficient coverage. Maybe the authors should double check for possible missed research works and update their results. After all they state that their goal is the totality of evidence (line 65).
The inclusion of 142 articles is too large. Usually, SRs end up with ~40-50 articles or less if the topic is novel.
Why only one reviewer evaluated the data? Usually, two members are employed to avoid any bias. This is commented on in limitations but not explained sufficiently.
What is the difference between - and ? in tables? Insufficient evidence is more or less synonym to uncertain evidence so till further evidence is available the em dash should be kept, right?
I think 2-4 sections are somehow overloading the manuscript so the authors should find a way to avoid it and keep a format that is suited for SRs (abstract, introduction, methods, results, discussion, and references)
What is missing and is confusing with the structure of the manuscript is the number and the type of data (in vitro/in vivo/clinical). This is also explained in limitations, but I think the statement that they can be extrapolated to clinical observations unless demonstrated otherwise is somehow overreaching (?). This is also a major issue of the SR. What is the actual outcome with sufficient evidence? What is only observed in vitro, and caution should be followed in PK estimations? what data are evident as in vivo =that can be used following a bottom-up approach.
As the authors conclude sparse and conflicting evidence exists and additional clinical PK data are needed. So, what is the contribution of their SR? If it was a narrative review, it could have stated this but in their work they aimed as stated to clarify in an extend these sparse evidence.
Comments on the Quality of English LanguageNone
Reviewer 2 Report
Comments and Suggestions for Authors
The authors of the manuscript “Review of Changes in Phase II Drug-Metabolizing 2 Enzyme and Drug Transporter Expression During Pregnancy” have presented a review on the modulation of drug-metabolizing enzymes and transporters during pregnancy. The literature review is extensive and impressively detailed. The results are described not only for the pregnancy but also for the placenta and after the delivery. This is a great review on an interesting topic. I was not able to find any additional literature that I considered to be included. The manuscript can be published in the present form.
Reviewer 3 Report
Comments and Suggestions for Authors
Comments for the authors: In this article, the authors collect the literature data (during the period of 26 May 2023 to 10 July 2023) about the changes in Phase II Drug-Metabolizing 2 Enzyme and Drug Transporter Expression During Pregnancy.
This study is well-designed, enjoyable, and contains novelties.
Minor Remark: - In my opinion, the “2.1. Placental Anatomy” part in the manuscript is unnecessary. I recommend skipping this.
Reviewer 4 Report
Comments and Suggestions for Authors
Manuscript titled “A Systematic Review of Changes in Phase II Drug-Metabolizing Enzyme and Drug Transporter Expression During Pregnancy “ is a very interesting review in the field of pregnancy and metabolism.
The overall structure is of good quality and easy to read. Methods and Results are clear and results corroborate the initial hypothesis of the authors.
Figures and Tables are of sufficient quality and easy to read as well as to understand to readers. However, manuscript need some improvements, specifically in Introduction and/or Discussion. Here the points:
1. Authors should add, in discussion, a proper description of the beneficial properties of nutraceuticals and their possible role in the affections of Drug-Metabolizing Enzyme and Drug Transporter Expression. For example, quercetin that is a beneficial nutraceutical with cardioprotective properties that coul be very useful during pregnancy as antioxidant and anti-inflammatory molecule ( cite this work 10.1007/s10973-017-6135-5 ).
2. Moreover, authors should explain how quercetin could change metabolizing enzyme during pregnancy ( cite this : doi: 10.3892/mmr.2016.5616. )
Based on these improvements, the article could be suitable for publication in this journal.
Reviewer 5 Report
Comments and Suggestions for Authors
Dear authors
First, let me congratulate you for a complete work on this very specific topic as drug-metabolizing enzymes and transporters during pregnancy.
As you mention in the article, this topic has been poorly treated, in general in the bibliography. Obviously is a very delicate and complex topic to be studied, which makes the review more important.
Experimental planning and summarizing tables are very useful and can be considered an important information source
Many thanks.
Round 2
Reviewer 1 Report
Comments and Suggestions for Authors
Dear authors,
Thank you for your detailed responses to my comments. First and foremost, I would like to congratulate you on the thorough study of the subject and your dedication to providing a comprehensive research analysis. I also believe that your work is detailed and would be of great interest. I respect the comments made by other reviewers who may have more experience in this field than I do. However, in my opinion, this work stands as an excellent literature review and not a systematic review. Your responses have further reinforced my belief.
According to your response, it appears that the best tools and practices for systematic reviews were not followed (see also Syst Rev 12, 96 (2023). https://doi.org/10.1186/s13643-023-02255-9). Your work leans more towards a qualitative summary of all available evidence from MEDLINE. Cochrane, known as the gold standard for clinical systematic reviews, recommends a minimum search of PubMed, Embase, and the Cochrane Central Register of Controlled Trials.
Quoting from J Clin Epidemiol. 2022 Sep:149:154-164. doi: 10.1016/j.jclinepi.2022.05.022
"Searching ≥2 databases improves coverage and recall and decreases the risk of missing eligible studies"
Quoting from Syst Rev 6, 245 (2017). https://doi.org/10.1186/s13643-017-0644-y
"Optimal searches in systematic reviews should search at least Embase, MEDLINE, Web of Science, and Google Scholar as a minimum requirement to guarantee adequate and efficient coverage."
Similar information can be extracted from Trop Med Health 47, 46 (2019). https://doi.org/10.1186/s41182-019-0165-6
Quoting from this "According to the Cochrane guidance, two reviewers are a must to do this step, but as for beginners and junior researchers, this might be tiresome; thus, we propose based on our experience that at least three reviewers should work independently". You only had one student.
This indicates that the timeline can extend from months to years, and often systematic reviews are conducted under a registered protocol to ensure transparency. The structure of a systematic review is highly specific with pre-specified eligibility criteria, a well-defined search strategy, assessment of the validity of findings, presentation and interpretation of results prior to the reference list. The 2-4 section structure is commonly found in literature reviews.
Systematic reviews represent 50% of the evidence-based hierarchy, with meta-analysis studies constituting the other 50%. Can we perform a meta-analysis in this work? More importantly, does this work meet the criteria of a systematic review? Can it provide regulatory bodies with robust information for regulatory reports?
In my scientific opinion, this work stands as an excellent literature review of the available data on Phase II drug-metabolizing enzyme and drug transporter expression during pregnancy. However, I cannot recommend it as a systematic review due to the deviation from best practices. I believe as a SR should be rejected. I will leave it with two options for you:
1) Edit the document to represent a literature review modulating also similar phrases.
2) Re-submit the work as a systematic review that adheres to best practices, incorporating the comments made above.
Comments on the Quality of English Language-
Author Response
Please see the attached letter

Round 3
Reviewer 1 Report
Comments and Suggestions for Authors
The authors have incorporated the proposed revisions into their text. The manuscript can now be further processed.